# Effects of Sleep on the Academic Performance of Children with Attention Deficit and Hyperactivity Disorder

**DOI:** 10.3390/brainsci11010097

**Published:** 2021-01-13

**Authors:** Lorena Villalba-Heredia, Celestino Rodríguez, Zaira Santana, Débora Areces, Antonio Méndez-Giménez

**Affiliations:** 1Faculty of Health Sciences, University of Zaragoza, 50009 Zaragoza, Spain; lvillalbaheredia@unizar.es; 2Faculty of Psychology, University of Oviedo, 33003 Asturias, Spain; arecesdebora@uniovi.es; 3Psicotogether, La Paloma Hospital, 35005 Las Palmas de Gran Canaria, Spain; zaira@psicotogether.com; 4Faculty of Education, University of Oviedo, 33005 Asturias, Spain; mendezantonio@uniovi.es

**Keywords:** ADHD presentations, sleep habits, accelerometry, primary education

## Abstract

Attention deficit and hyperactivity disorder (ADHD) is commonly associated with disordered or disturbed sleep and the association of sleep problems with ADHD is complex and multidirectional. The purpose of this study was to analyze the relationship between sleep and academic performance, comparing children with ADHD and a control group without ADHD. Academic performance in Spanish, mathematics, and a foreign language (English) was evaluated. Different presentations of ADHD were considered as well as the potential difference between weekday and weekend sleep habits. The sample consisted of 75 children aged 6–12 in primary education. Accelerometry was used to study sleep, and school grades were used to gather information about academic performance. The results showed that ADHD influenced the amount of sleep during weekends, the time getting up at the weekends, weekday sleep efficiency, as well as academic performance. Given the effects that were seen in the variables linked to the weekend, it is necessary to consider a longitudinal design with which to determine if there is a cause and effect relationship.

## 1. Introduction

Attention deficit and hyperactivity disorder (ADHD) is a neurodevelopmental disorder which begins in childhood and often persists into adolescence and adult life. It is a symptomatic triad, characterized by a series of symptoms or patterns of more or less stable indicators, such as hyperactivity, impulsivity, and attention deficit, which cause deterioration in family, academic, social, and work-related functioning [1]. ADHD has a genetic basis and is chronic in nature. It has been considered a neurobiological disorder given the belief that there may be altered functioning of some areas of the brain. This belief is based on studies such as Shaw et al. [2], which showed that there is a delay in brain maturation in ADHD, supported by the anatomical characteristics of the brain (thickness of the gray matter of the brain). One must avoid making the mistake of thinking that this always manifests in the same way and that all children have the same characteristics given the heterogeneity of ADHD symptomatology. Currently, there are different presentations of ADHD depending on the characteristics or common features that are exhibited. The Diagnostic and Statistical Manual of Mental Disorders, Fifth Edition (DSM-5) describes three presentations of ADHD [1,3]. ADHD-I is the most common presentation, and is associated with a lack of attention, and greater presence of symptoms of inattention, and difficulty in organizing and planning tasks. The hyperactive/impulsive presentation, also known as ADHD-H/I, is associated with problems of self-regulation of behavior. Finally, the combined presentation, or ADHD-C, presents a combination of the above (inattention and hyperactivity), presenting more difficulties in visuospatial working memory than ADHD-I presentation.

ADHD is one of the most common mental health problems in the pediatric population, affecting 5–7% of the school population, and is more frequent in boys than in girls [4]. At the school level, in a class of 30 students, this corresponds to one or two students with ADHD per classroom. Its prevalence in the adult population is estimated to be between 3% and 4% [5].

Another aspect to consider is comorbidity, the existence of two different diseases that appear simultaneously due to different biological processes, creating two different sets of symptoms, or additional symptoms in the course of a disorder that makes them different [6]. There are multiple comorbid disorders associated with ADHD, such that significant comorbidity with ADHD often hinders its investigation and detection. The most frequent comorbid disorders are affective problems, anxiety, aggressive impulsive behavior, challenging negativist disorder and oppositional defiant attitude, chronic tics or Tourette’s syndrome, mood swings with or without bipolar syndrome, language and communication difficulties, motor coordination disorders, a tendency to use alcohol and other drugs, learning difficulties, school failure, driving problems, and antisocial behavior [7].

ADHD also influences learning behavior, as well as adaptive processes in the school environment. This can affect the relationship with teachers, classmates, and the family. It is common for people who suffer from this disorder to have sleep problems, and the association between sleep disorders and ADHD is very complex. Sleep disorders usually appear during childhood and are characterized by problems of sleep consolidation, night terrors, and sleepwalking, among others [8].

### 1.1. The Need for Sleep

Sleep is a recurring, restorative act that humans spend a third of their lives doing, it has many definitions depending on the science examining it. It was not until 1929 that sleep began to be studied as a physiological activity [9]. Mesarwi, Polak, Jun, and Polotsky [10] agree that sleep is a recurrent, involuntary physiological state, where sensory activity and the level of consciousness are reduced. Tynjala, Kannas, and Vällmaa [11] conducted a study examining the sleep habits of young Europeans, in 11 European Union countries. Their results indicated that 11–12-year-old children spent 9–10 h a day in bed. According to Hirshkowitz et al. [12], experts recommend 9–11 h of real sleep for school-aged children (6–13). It should be noted that there is a difference between actual sleep time and the time spent in bed. Short et al. [13] compared actual time spent sleeping to current sleep recommendations and showed that most children studied had sleep times that did not meet the recommendations of the National Sleep Foundation [14] or the American Academy of Sleep Medicine.

The quantification of sleep is as important as it is complex, and there are many ways to quantify it. As Hamze, Souza, and Chianca [15] summarized, in recent years, different objective measurement methods suitable for field studies have been developed (such as polysomnography and accelerometry) in contrast to subjective methods such as questionnaires and sleep diaries. The use of devices and instruments such as frequency monitors and accelerometers has become generalized, allowing individual objective measurements. Although they are very accurate, they are expensive and there are other problems that make it difficult to use them in studies with large populations.

With respect to subjective methods, the questionnaire is one of the most widely-used techniques for estimating sleep levels at the population level. Despite the tendency of respondents to misreport their actual sleep levels, questionnaires or validated sleep measurement scales are considered suitable tools for measuring this variable. One example of a questionnaire designed to measure sleep in children and adolescents is the Brief Infant Sleep Questionnaire (BISQ), which calculates time taken to fall asleep, duration of sleep, and night awakenings during the previous week [16]. Another example is the Children’s Sleep Habits Questionnaire (CSHQ), which consists of 33 items to identify disorders in sleep habits, latency, time getting up, total sleep time, and quality of sleep [17].

One of the limitations in comparing data between different studies is the lack of a common methodology allowing replication and objective discussion of results. However, questionnaires allow the population being studied to be positioned in relation to reference values. Studies such as that by Sadeh [18] established that accelerometry can analyze students’ sleep levels during a week.

This study provides objective, sex- and age-segregated information about the duration and quality of sleep of students with ADHD. Accelerometers allow us to objectively observe what a night’s rest is like in this population. Accelerometers have a flexible band to place them on the hip, which has been shown to be the region providing the most reliable data using this technique [19].

### 1.2. Sleep and ADHD

ADHD is commonly associated with disordered or disturbed sleep and the association of sleep problems with ADHD is complex and multidirectional [20,21,22,23,24,25]. High nocturnal activity and disordered sleep are defining characteristics of ADHD [24,26,27,28]. De Crescenzo et al. [29] showed evidence for higher mean activity in ADHD compared to children without ADHD, as expected. In the same line, another study [30] reported that there were no differences in sleep between the presentations of ADHD [30]. Studies using subjective measures such as questionnaires found children with ADHD to have more sleep disturbances than non-ADHD children, while studies using objective measures such as polysomnography or actigraphy have produced inconsistent results [30,31].

Actigraphy, as a measure of mean activity and sleep patterns in children with ADHD treated with psychostimulants such as methylphenidate, might be a valuable tool for prescribing clinicians, who must balance efficacy in hyperactivity against adverse effects on sleep that methylphenidate may cause [32]. In this regard, De Crescenzo et al. [33] indicated that data comparing ADHD activity measures against healthy controls had never been pooled, with and without medication, against healthy children, which does not allow for a reliable estimate of the difference between a child with ADHD and a child with a sleep disorder.

### 1.3. Associations between Academic Performance and Sleep

According to Kelman [34] and Becker, Langberg, Eadeh, Isaacson, and Bourchtein [35], there is a relationship between sleep duration and adolescent students’ academic performance. Short sleep times were associated with reduced academic performance. In contrast, Unalan, Orturk, Ismaliogullari, Akgul, and Aksu [36] established that sleep duration was inversely related to student performance, with longer sleep being related to poorer academic performance. Dewald, Meijer, Oort, Kerkhof, and Bögels [37] determined the existence of a significant relationship between sleep and performance in children and adolescents, the less often they woke up in the night, the better their performance, and the longer they slept, the better their performance. These results were more significant in adolescents than in infants. In a study analyzing sleep patterns with accelerometry and evaluating the academic results of students aged between 8 and 10, no significant differences between the variables were seen, suggesting that sleep time had no effect on academic performance [38].

ADHD is associated with a significant motor limb load as well as irregular sleep [27]. This is reflected in studies by Sangal et al. [39], Gruber et al. [40], and Cortese, Faraone, Konofal, and Lacendreux [32] in which there were differences in the quality and quantity of sleep between a control group and a group with ADHD. The results showed children with ADHD to be significantly worse than controls in most subjective and some objective measures of sleep. However, not all studies have found differences between children with ADHD and control groups. For example, after analyzing the association between sleep patterns and ADHD, Goodlin-Jones, Waters, and Anders [41] found no difference between the two groups.

Given that the literature on this subject is insufficient and that there are discrepancies in the results, it is an appropriate topic of study. The main objective of this study is to analyze the degree to which sleep quality, quantity, and habits (measured objectively through accelerometry) influence academic performance in Spanish, mathematics, and English in students with ADHD, examining the different presentations of ADHD, a control group, and sleep during weekdays and at weekends.

## 2. Materials and Methods

### 2.1. Participants

The sample consisted of 75 children aged between 6 and 12, with a mean age of 9.47 years (*SD* = 1.73), from both state-funded and independent primary schools in the north of Spain. The control group consisted of 28 children, and the experimental group of 47 children with ADHD. It is important to emphasize that those participants in the ADHD group who were having pharmacological treatment continued their treatment during the whole evaluation process. There were no significant differences between groups according to age (*p* = 0.231), intelligence quotient (IQ) (*p* = 0.941), or body mass index (BMI) (*p* = 0.161). The groups were not equivalent by sex since ADHD is more frequent in boys than girls (*χ*^2^ (1) = 4.587; *p* = 0.032). Table 1 shows the main descriptive statistics of the sample as a whole for the variables in this study.

#### Exclusion/Inclusion Criteria

The ADHD sample was identified by mental-health professionals (typically one or more psychiatrist–neurologists) on the basis of the following criteria: (a) clinical diagnosis of ADHD according to the Diagnostic and Statistical Manual of Mental Disorders-5 [1]; (b) symptoms lasting more than 1 year; (c) the problem beginning before the age of 7; and, (d) no associated disorders. Participants who presented with a cognitive deficit, Asperger’s syndrome, Tourette’s syndrome, or extensive anxiety depressive disorders were excluded from the study. To confirm the diagnosis and rule out other associated disorders, all parents underwent the semi-structured interview Diagnostic Interview Schedule for Children DISC-IV [42]. The children were also administered a Spanish version of the Cattell *g* test of general intelligence [43] to evaluate the presence of specific (or other) cognitive deficits. As part of their diagnosis, the children were identified as exhibiting one of three ADHD presentations—inattention, hyperactivity/impulsiveness, or the combined form. Nearly all of these children (94%) had been prescribed medication to control their ADHD symptoms. Children with an IQ below 75 or above 130 were excluded from the study.

The socioeconomic levels of the participants’ families were medium to low, and the families mostly had low levels of educational qualifications (elementary education). The sample as a whole was initially divided into two groups: the non-ADHD group, made up of students from a state-funded school and the ADHD group, comprising students diagnosed with ADHD, from both state-funded and independent schools.

In order to identify the groups, and to see the relationship between the variables and the different presentations of ADHD, four new groups were defined. The “Inattention” group, was made up of students scoring above the 90th percentile in I (inattention) items in the parent Assessment of Attention Deficit Hyperactivity Disorder test [42]. The “Hyperactive/Impulsive” group comprised students scoring above the 90th percentile in the H/I (Hyperactivity/Impulsivity) items in the parental ADHD assessment. The “Combined” group was made up of students scoring above the 90th percentile in the sum of the previous items (Hyperactivity/Impulsivity and Inattention) in the parental ADHD assessment. Finally, the “Control group” consisted of students who did not have an ADHD diagnosis. In order to confirm the absence of ADHD symptoms, the same scale was used as for the experimental group (the Parent Assessment of Attention Deficit-Hyperactivity Disorder test [42]). Using this scale ensured that the Control group was made up of children without any ADHD-related symptoms. Participants who recorded four or more days without using the ActiGraph-GT3X device were removed from the study.

### 2.2. Design and Procedure

This study used a quantitative methodology, and a descriptive, correlational cross-sectional design (ex-post-facto). The sampling method chosen was non-probabilistic casual or accessibility sampling for both groups (control and experimental group). The research group met with the children’s families to inform them about the study and to answer any questions they had about the project. Once the meeting was finished, the parents who consented to their children’s participation in the study signed their informed consent. The study was conducted in accordance with The Code of Ethics of the World Medical Association (Declaration of Helsinki), which reflects the ethical principles for research involving humans [43]. The study was approved by the Ethical Committee of the Principality of Asturias (reference: CPMP/ICH/135/95, code: TDAH-Oviedo) and all procedures were carried out in compliance with relevant laws and institutional guidelines. Data were collected from children, schools, and parents. Parents were informed about the study by the school authorities, and were assured of the data confidentiality policy before signing the informed consent document.

### 2.3. Instruments and Measures

We used a variety of information collection tools depending on the nature of the variables in the study.

#### 2.3.1. Independent Variables

Sex, age, socioeconomic level, school year, and type of school. This information was provided by the schools and parents.Evaluation scale of Attention Deficit and Hyperactivity Disorder (EDAH). The ADHD measure from Farré and Narbona [44] measures the main features of ADHD and behavioral disorders that may coexist with it. It is intended for the evaluation of children aged between 6 and 12. It is applied individually, lasting approximately 5 to 10 min. The scale has 20 items, with a scale ranging from 0 to 3. This questionnaire demonstrated high reliability (*α* = 0.97).Intelligence Quotient (IQ). Cattell’s *g*-factor [45], Scale 2-Forms A and B consists of 4 subtests, devised for children aged 8 to 18. The *g*-factor tests are a very useful instrument for assessing intelligence, presenting reliability indices that are estimated above 0.85. It is a nonverbal test, applied individually, and is very short. The direct score in each subtest is the number of correct answers. The total score is the sum of the four subtest scores. For the conversion to IQ, only the total score is used, with a maximum score of 46 points.

#### 2.3.2. Dependent Variables

Sleep. The children’s sleep duration and quality was objectively quantified using ActiGraph-GT3X accelerometry (ActiGraphTM, LLC, Fort Walton Beach, FL, USA), collecting data using a triaxial function, 10 s sampling times, and 30 Hz. The children wore the accelerometers twenty-four hours a day, seven days a week, taking them off only during showers or water-based activities (such as swimming or diving) as the devices are not waterproof. The use of these accelerometers allowed us to objectively quantify not only the efficiency but also the duration of actual sleep, the time spent in bed, as well as the time going to bed and getting up. Accelerometry has been established as a valid instrument to estimate sleep [18]. According to the study by Medrano [46], Spanish children’s weekday bedtimes are between 10 p.m. and 11 p.m. This is in line with the study by Yokomaku et al. [47] which established 11 p.m. as the weekday bedtime [48]. This led to us establishing 11 p.m. as a cut-off point for weekdays and 11:30 p.m. for weekends. During weekdays, the time children get up is affected by going to school, getting up before 9 a.m., so that 8 a.m. was taken as a reference point for weekdays, and 9 a.m. for weekends [46,48].Academic performance. Children’s academic performance was measured using their grades in mathematics, Spanish, and English during the 2018/19 school year. The grades were provided by the schools for the control group and by the parents of the experimental group. Academic grades allow comparisons with other studies that have used academic performance as one of their main constructs. It is worth emphasizing that Spain’s education system sets out precise evaluation criteria by participant matter and school year, allowing the use of normative evaluation systems. Moreover, the academic performance variables were analyzed in line with previous international studies [36,37,38].

#### 2.3.3. Data Analysis

The data collected was stored and processed using an SPSS 22.0 database. The level of statistical significance was established at a value of *p* < 0.05. To analyze the existence of statistically significant differences between the groups, we performed a univariate analysis of covariance (ANCOVA), with the membership group (ADHD and control) as independent variables and academic performance and sleep variables as dependent variables.

In order to identify the groups between which there were differences, we performed multiple comparisons with the Sheffé statistic. We carried out simultaneous measurement of the variables; academic variables were included first, followed by control variables (sex, and school year), then finally the sleep variables.

## 3. Results

### 3.1. Preliminary Analysis

Asymmetry and kurtosis values suggested a normal distribution (Table 2). We followed the guidelines from Gravetter and Walnau [49] who propose that the maximum in both cases be ±2 for parametric analyses. We also calculated Pearson correlations for all variables analyzed (variables related to academic performance and sleep variables), producing significant correlations as Table 2 shows.

### 3.2. Relationship between Sleep Efficiency, Time Spent in Bed, Duration of Sleep and ADHD

The ANCOVA analyses with the variables sex and age as covariates (Table 3) indicated statistically significant differences (*p* < 0.001) for the academic performance variables by group (Table 3). ADHD explained 33.0% of the variance in academic performance in Spanish, compared with 30.6% and 19.1% in mathematics and English, respectively. Regarding sex covariate, it is observed a tendency to present higher results in men than women.

The ANCOVA analyses (Table 3) did not indicate statistically significant differences between groups for the variables of sleep efficiency or time spent in bed, or for weekday and weekend time spent sleeping. ADHD demonstrated no influence on weekday sleep efficiency or duration. It had somewhat more influence at the weekend, in which ADHD explained 5.6% of the variance of weekend time spent sleeping.

### 3.3. Relationship between Bedtime and Waking Habits and ADHD

No statistically significant differences were found between groups in the variables related to bedtimes during the weekend or getting up on weekdays or during the weekend. There were statistically significant differences in weekday bedtimes. Thirty-two percent of children went to bed before 11 p.m., while the remaining 68% did so after 11 p.m., with differences between the ADHD group (*ES* = 0.103; *Z* = −3.094; *p* > 0.001) and the group without ADHD (*ES* = 0.103; *Z* = −4.332; *p* < 0.001). There were no statistically significant differences in weekend bedtimes between the groups (*ES* = 0.134; *Z* = −3.207; *p* > 0.001). During weekends, only 33.3% of children got up before 9 a.m., compared to 66.7% who got up later.

### 3.4. Sleep Parameters and the Different Presentations of ADHD

The ANCOVA analyses indicated statistically significant differences in the academic performance variables by group. The inter-subject effects are shown in Table 4. ADHD in any of its presentations explained 38.0% of the variance of academic performance in Spanish, while in mathematics and English, it explained 31.0% and 23.0%, respectively.

The univariate tests in the ANCOVA analyses showed statistically significant differences in weekday sleep duration and weekday sleep efficiency depending on the group. The presence of ADHD explained 19% of the variance of weekday sleep efficiency. There were no significant differences by group in the other weekday variables.

The presence of any of the three presentations of ADHD had no significant influence on the variance of weekend bedtime, weekend sleep efficiency, or actual weekend sleep time.

Deeper analysis of academic performance using Scheffé’s test post hoc for the various ADHD manifestations showed that the Control group scored higher marks in Spanish than either the H/I (*p* = 0.017, *MD* = 1.71, *d* = 4.66) or the C (*p* < 0.001, *MD* = 2.52, *d* = 9.47) groups.

Similar differences were seen in the Control group’s performance in mathematics compared to the H/I group (*p* = 0.028, *MD* = −1.97, *d* = −4.34) and the C group (*p* < 0.001, *MD* = −2.21, *d* = −6.79). There was also a difference in the means of the Control group and the C group in English *(p* = 0.001, *MD* = −1.95, *d* = −6.08).

Statistically significant differences were also found in mean weekday sleep efficiency between the H/I and C groups (*p* = 0.007, *MD* = 3.25, *d* = 5.13). No differences between groups were found for any of the other target variables.

### 3.5. Relationship between Weekday Bedtime Routine and Different Presentations of ADHD

Testing univariate models in the ANOVA analysis showed statistically significant differences in the interaction of the different presentations of ADHD in academic performance in different participants and weekday bedtime as shown in Table 5, with *n* = 37 with weekday bedtime before 11:30 p.m. and *n* = 38 after 11:30 p.m. Manifestation of ADHD presentations explained 21.1% of the variance of academic performance in Spanish in relation to weekday bedtime, while the influence of bedtime on academic performance in mathematics was 28.4% of the variance. There were no statistically significant differences in academic performance in English in relation to the weekday bedtime variable. However, when we analyzed each subgroup (<11 p.m. vs. ≥11 p.m.), the combined presentation of ADHD demonstrated significant differences in comparison with its Control group, with the latter having better results.

Specifically, there were statistically significant differences for the independent variable weekday bedtime between the mean academic performance in Spanish from the Control group and the H/I group (*p* = 0.001). The Control group scored 1.707 (*d* = 3.33) points more than the H/I group, these differences were also seen in mathematics (*p* < 0.001), with the Control group mean 1.971 (*d* = 3.09) points higher than the H/I group.

There were statistically significant differences between the mean grades of the Combined and Control groups in Spanish and mathematics (*p* = 0.001, *p* = 0.000 respectively), with the Control group averaging 2.518 (*d* = 6.77) and 2.214 (*d* = 4.78) points higher than the Combined group in Spanish and mathematics respectively, in English the Control group scored 1.946 (*d* = 4.29) higher than the Combined group.

### 3.6. Relationship between Weekend Bedtime and Getting Up Routine and Different Presentations of ADHD

Table 6 shows the ANOVA results of the interaction between weekend bedtimes, the different ADHD presentations, and academic performance.

Statistically significant differences were found in the mean scores in the three participants between the H/I ADHD group and the Control group, as well as between the Combined ADHD group and the Control group weekend bedtime before 11:30 p.m. There were also statistically significant differences for the variable weekend bedtime in the mean scores in mathematics (*p* < 0.000), with the Control group scoring 2.467 points higher than the H/I group. Similarly, in English (*p* = 0.001), the Control group average was 3.625 points higher than the H/I group.

The Control group was significantly different to the Combined group in all three participants of Spanish, mathematics, and English (*p* = 0.000, *p* = 0.000, and *p* = 0.033), with average scores that were 2.553, 2.462, and 2.058 points higher in the Control group for each participant, respectively.

The H/I group was significantly different to the I group in Spanish and mathematics (*p* = 0.023, *p* = 0.000), with average scores that were 2.625 and 4.5 points higher in the H/I group for each participant respectively than I group. If compare it to weekend bedtime after 11:30 p.m., the Control group was significantly different to the Combined group in Spanish and mathematics (*p* = 0.000 and *p* = 0.017), with average scores that were 2.217 and 1.8 points higher in the Control group for each participant respectively than C group. The H/I group was significantly different to the Combined group in Spanish and mathematics (*p* = 0.000, *p* = 0.034), with average scores that were 2.467 and 2.133 points higher in the H/I group for each participant respectively than C group. Table 7 show post hoc multiple comparison.

No statistically significant differences were found in relation to the time getting up during the week or at the weekend and the different ADHD presentations.

## 4. Discussion

The results suggest that the weekday rest patterns of students with ADHD, and the Control group, did not affect academic results in mathematics, Spanish, and English. In contrast, bedtime and getting up routines did affect the academic performance of students with ADHD, in line with Garcia et al. [50], who reported how academic performance decreases with executive problems. This indicates that a shorter time in bed, and therefore less real sleep, negatively affects academic performance. Kelman [34], Dewald et al. [37], and Quevedo-Blasco [51] also found a directly proportional relationship between sleep duration, sleep quality, and school performance. Academic performance decreases as sleep quality and duration decline. On the other hand, Unalan et al. [36] reported an inversely proportional relationship between sleep duration and academic performance, with improved performance as real sleep time decreased.

Our results show that there were no differences in sleep duration or quality during weekdays, in part because of school timetables, which keep sleep schedules within a normalized range during the weekdays both in children with ADHD and without ADHD. These findings are in line with what was observed by San gal et al. [39], Gruber et al. [40], and Cortese et al. [32]. However, there were differences between the groups in the duration of real sleep at the weekend, with the ADHD group exhibiting less real sleep time than the Control group, something which is very common in ADHD as recorded by Corkum et al. [52] and Sung et al. [21], who stated that sleep problems were frequent in 50–70% of cases of children with ADHD.

However, despite the group presenting ADHD with Hyperactivity/Impulsivity having less efficient sleep during the week than the group with Combined ADHD, they demonstrated higher academic performance. On the other hand, sleep during the week did not play an important role in the performance of students with or without ADHD. These results are consistent with those from Erath et al. [38], Goodlin-Jones et al. [41], and Eliasson, King, and Gould [53], in which the efficiency and duration of sleep had no effect on academic performance. Conversely, authors such as Short et al. [13] established that longer sleep durations are associated with better cognitive functioning, and consequently better academic performance.

We found an association between the habits of when primary students got up and their academic performance, in line with what authors such as Trockel et al. [54], and Wolfson and Carskadon [55] reported. Those authors noted a relationship between the time participants got up and academic performance; performance fell the later participants got up. Additionally, an indirect relationship was found between weekday and weekend bedtime and academic performance in the different presentations of ADHD, along the lines outlined by authors such as BaHamman et al. [56] and Cecchini-Estrada et al. [57], who demonstrated positive effects between bedtime and academic performance.

### Future Perspectives and Limitations

Due to the clinical and educational implications of the results, there is a need to raise awareness among parents and professionals about creating sleep habit programs for children with ADHD. The results of this study, as well as the existing literature, highlight the need to continue working on this topic with tools that allow us to quantify sleep accurately. Given the discrepancies between the studies carried out so far, whether sleep has a direct relationship to academic performance or not, it would be interesting to examine this relationship more deeply, including the influence of daytime sleep, as well as to examine the normalization of sleep under treatment. Another limitation of the present study is associated to the limited composition of the group considering the variable “bedtime breakdown”. In this sense, it is necessary to increase the sample in order to compare the groups.

## 5. Conclusions

This work focused on establishing relationships without determining cause and effect. Our results showed a relationship between weekday and weekend bedtimes, which correlated with academic performance in Spanish, mathematics, and English in children with ADHD and children without this disorder, as well as with the different presentations of ADHD. The study also showed the influence of the efficiency of weekdays sleep on the academic performance of the different presentations of ADHD. In contrast, it did not show a relationship between the duration of sleep, the time spent in bed, or weekend getting up time and children’s academic performance. ADHD influences academic performance, as well as the duration of real sleep during the weekend.

## Figures and Tables

**Table 1 brainsci-11-00097-t001:** Sample characteristics. Description of the variables.

Variables	ADHD (*n* = 47)	Non-ADHD (*n* = 28)	Total (*n* = 75)
*Sex*			
Boy	33 (70.2%)	15 (53.6%)	48 (64%)
Girl	14 (29.8%)	13 (46.4%)	27 (36%)
*ADHD Presentation*			
Inattentive	9 (19.1%)	-	9 (12%)
Hyperactive/Impulsive	10 (21.3%)	-	10 (13.4%)
Combined	28 (59.6%)	-	28 (37.3%)
Control Group	-	28 (100%)	28 (37.3%)
*Age (years)*	9.62 (1.53)	9.21 (1.95)	9.47 (1.70)
*IQ*	101.85 (13.41)	101.94 (10.61)	101.89 (12.01)
*School Years*			
1st and 2nd years	9 (19.1%)	7 (25%)	16 (21.3%)
3rd and 4th years	20 (42.6%)	10 (35.7%)	30 (40%)
5th and 6th years	18 (38.3%)	11 (39.3%)	29 (38.7%)
*Medication*			
Yes	30 (63.8%)	-	30 (40%)
No	17 (36.2%)	28 (100%)	45 (60%)
*Schools*			
State funded	26 (55.3%)	28 (100%)	54 (72%)
Independent	21 (44.7%)	-	21 (28%)
*Mother’s education*			
Secondary and above	17 (36.2%)	19 (67.9%)	36 (48%)
Elementary	30 (63.8%)	9 (32.1%)	39 (52%)
*Father’s education*			
Secondary and above	15 (31.9%)	11 (42.3%)	26 (34.7%)
Elementary	29 (68.1%)	15 (57.7%)	44 (65.3%)

Note. Continuous variables are expressed as mean (standard deviation), categorical variables are shown as frequency (percentage).

**Table 2 brainsci-11-00097-t002:** Descriptive statistics and Pearson correlation matrix.

Variables	1	2	3	4	5	6	7	8	9
1	-	0.77 **	0.64 **	−0.11	−0.01	−0.05	−0.14	−0.15	−0.11
2		-	0.55 **	0.01	0.05	0.04	0.03	−0.22	−0.16
3			-	0.01	0.15	0.13	−0.14	−0.11	−0.09
4				-	−0.01	0.22	0.34 **	−0.05	0.01
5					-	0.97 **	0.05	0.32 **	0.41 **
6						-	0.13	0.31 **	0.41 **
7							-	−0.17	−0.04
8								-	0.92 **
9									-
M	6.53	6.31	6.89	95.88	571.92	547.20	95.49	591.48	561.72
SD	1.77	1.97	1.89	2.60	64.25	63.10	3.17	94.02	78.99
Asymmetry	0.09	0.03	0.30	−1.09	0.64	0.63	−1.03	0.66	−0.09
Kurtosis	−0.99	−0.52	−0.95	1.43	2.50	2.49	0.87	2.20	1.41

Note: 1. Academic performance Spanish; 2. Academic performance mathematics; 3. Academic performance English; 4. Weekday sleep efficiency; 5. Weekday time spent in bed; 6. Weekday time spent sleeping; 7. Weekend sleep efficiency; 8. Weekend time spent in bed; 9. Weekend time spent sleeping. ** *p* < 0.01

**Table 3 brainsci-11-00097-t003:** Inter-subject effects of Attention deficit and hyperactivity disorder (ADHD) on dependent variables.

Variables	ADHD *M* (*SD*)	Non-ADHD *M (SD)*	*F*	*p*	*ηp* ^2^	SEX *ηp*^2^	AGE *ηp*^2^
**Academic performance**							
Spanish performance	5.75 (1.49)	7.86 (1.38)	34.44	0.000	0.330	0.000 (0.926)	0.004 (0.592)
Math performance	5.56 (1.63)	7.57 (1.85)	30.88	0.000	0.306	0.054 * (0.048)	0.000 (0.920)
English performance	5.83 (1.69)	7.50 (1.75)	16.55	0.000	0.191	0.004 (0.580)	0.003 (0.674)
**Sleep variables**							
Weekday sleep efficiency	95.85 (2.91)	95.93 (2.00)	0.01	0.927	0.000	0.009 (0.420)	0.032 (0.132)
Weekday time spent in bed	571.47 (65.29)	572.67 (63.64)	0.06	0.808	0.001	0.003 (0.621)	0.008 (0.465)
Weekday time spent sleeping	546.81 (66.94)	547.85 (57.24)	0.05	0.823	0.001	0.008 (0.462)	0.001 (0.774)
Weekend sleep efficiency	96.01 (2.86)	94.62 (3.52)	2.58	0.113	0.036	0.017 (0.266)	0.005 (0.563)
Weekend time spent in bed	604.87 (103.97)	569.00 (70.52)	3.63	0.061	0.049	0.052 (0.052)	0.001 (0.806)
Weekend time spent sleeping	573.49 (85.46)	541.96 (63.36)	4.21	0.044	0.056	0.060 * (0.037)	0.000 (0.913)

Note: *M* = mean; *SD* = standard deviation; Sex *ηp*^2^ = the values without parentheses are the effect size; the values in parenthesis are *p* values. Age *ηp*^2^ = the values without parentheses are the effect size; the values in parenthesis are *p* values.* *p* < 0.05.

**Table 4 brainsci-11-00097-t004:** Inter-subject effects for the four groups in the dependent variables.

Variables	CG *M* (*SD*)	I *M* (*SD*)	H/I *M* (*SD*)	C *M* (*SD*)	*F*	*p*	*ηp* ^2^	SEX *ηp*^2^	AGE *ηp*^2^
**Academic performance**									
Spanish	7.86 (0.27)	6.56 (0.47)	6.15 (0.45)	5.34 (0.27)	13.94	0.000	0.38	0.070	0.339
Mathematics	7.57 (033)	6.17 (0.57)	5.60 (0.55)	5.36 (0.33)	10.10	0.000	0.31	0.008	0.361
English	7.50 (0.32)	6.78 (0.57)	5.75 (0.54)	5.55 (0.32)	6.91	0.000	0.23	0.018	0.091
**Sleep variables**									
Weekday sleep efficiency	95.93 (0.46)	95.59 (0.81)	93.51 (0.77)	96.76 (0.46)	5.26	0.003	0.19	0.005	0.058
Weekday time spent in bed	572.67 (12.10)	591.48 (21.34)	539.69 (20.25)	576.39 (12.10)	1.10	0.354	0.05	0.000	0.166
Weekday time spent sleeping	547.85 (11.68)	562.72 (20.60)	504.11 (19.54)	556.94 (11.68)	2.03	0.118	0.08	0.000	0.161
Weekend sleep efficiency	94.62 (0.58)	96.00 (1.03)	94.48 (0.98)	96.56 (0.58)	1.78	0.160	0.07	0.005	0.043
Weekend time spent in bed	569.00 (17.67)	597.00 (31.16)	579.23 (29.57)	616.56 (17.67)	1.88	0.141	0.08	0.051	0.019
Weekend time spent sleeping	541.97 (14.74)	572.56 (26.01)	543.78 (24.67)	584.41 (14.74)	2.37	0.078	0.09	0.063	0.014

Note: I = Inattention Presentation; H/I = Hyperactive/Impulsive Presentation; CG = Control group; C = Combined presentation. Sex *ηp*^2^ = the values provided are the effect size (partial eta squared); Age *ηp*^2^ = the values provided are the effect size (partial eta squared).

**Table 5 brainsci-11-00097-t005:** Inter-subject effects for the four groups on the bedtime routine on weekdays.

Academic Performance	Weekday Bedtime	*n*	CG *M* (*SD*)	*n*	I *M* (*SD*)	*n*	H/I *M* (*SD*)	*n*	C *M* (*SD*)	*F*	*p*	*ηp^2^*
Spanish	<11:30 p.m.	16	8.63 (0.97)	4	7.00 (1.63)	4	4.37 * (0.95)	13	5.69 * (1.03)	5.988	0.001	0.211
	≥11:30 p.m.	12	7.55 (1.43)	5	6.20 (1.48)	6	7.33 (1.03)	15	5.20 (1.32)			
Mathematics	<11:30 p.m.	16	8.00 (1.31)	4	7.50 (1.29)	4	3.00 * (1.41)	13	5.25 * (1.04)	8.846	0.000	0.284
	≥11:30 p.m.	12	7.40 (2.04)	5	5.10 * (1.67)	6	7.33 (0.52)	15	5.40 * (1.23)			
English	<11:30 p.m.	16	7.63 (1.69)	4	7.25 (1.71)	4	4.63 (1.11)	13	5.94 * (1.37)	1.365	0.284	0.058
	≥11:30 p.m.	12	7.45 (1.82)	5	6.40 (1.14)	6	6.50 (1.76)	15	5.40 * (1.82)			

Note: I = Inattention Presentation; H/I = Hyperactive/Impulsive Presentation; CG = Control group; C = Combined presentation; * = Dif CG.

**Table 6 brainsci-11-00097-t006:** Inter-subject effects for the four groups on weekend bedtime routine.

Academic Performance	Weekend Bedtime	CG *M* (*SD*)	I *M* (*SD*)	H/I *M* (*SD*)	C *M* (*SD*)	*F*	*p*	*ηp* ^2^
Spanish	<11:30 p.m.	8.44 (1.21)	7.00 (1.63)	4.37 (0.95)	5.89 (1.19)	18.782	0.000	0.457
	≥11:30 p.m.	7.08 (1.24)	6.20 (1.48)	7.33 (1.03)	4.87 (1.13)			
Mathematics	<11:30 p.m.	8.00 (1.67)	7.50 (1.29)	3.00 (1.41)	5.54 (1.13)	11.655	0.000	0.343
	≥11:30 p.m.	7.00 (2.00)	5.10 (1.67)	7.33 (0.52)	5.20 (1.21)			
English	<11:30 p.m.	8.25 (1.48)	7.25 (1.71)	4.63 (1.11)	6.19 (1.77)	7.186	0.000	0.243
	≥11:30 p.m.	6.50 (1.62)	6.40 (1.14)	6.50 (1.76)	5.00 (1.46)			

Note: I = Inattention Presentation; H/I = Hyperactive/Impulsive Presentation; CG = Control group; C = Combined presentation.

**Table 7 brainsci-11-00097-t007:** Post hoc multiple comparisons for the four groups with Bonferroni corrections.

Academic Performance	Weekend Bedtime	I vs. CG	H/I vs. CG	C vs. CG	I vs. H/I	C vs. I	C vs. H/I
Spanish	<11:30 p.m.	−1.44 (−1)	−4.06 *** (−3.74)	−2.55 *** (−2.15)	2.63 * (1.97)	−1.12 (−0.78)	1.51 (1.41)
	≥11:30 p.m.	−0.88 (−0.65)	0.25 (0.22)	−2.22 *** (−1.86)	−1.13 (−0.89)	−1.33 (−1.01)	−2.47 *** (2.28)
Mathematics	<11:30 p.m.	−0.5 (−0.34)	−5 (−3.24)	−2.46 *** (−1.73)	4.5 *** (3.33)	−1.96 (−1.62)	2.06 * (1.99)
	≥11:30 p.m.	−1.9 (−1.03)	0.33 (0.23)	−1.8 * (−1.09)	−2.23 (−1.8)	0.1 (0.07)	−2.13 (−2.29)
English	<11:30 p.m.	−1 (−0.63)	−3.63 *** (−2.77)	−2.06 * (−1.26)	2.63 (1.82)	−1.06 (−.61)	1.57 (1.06)
	≥11:30 p.m.	−0.1 (−0.07)	0.00 (0)	−1.5 (−0.97)	−0.1 (−0.07)	−1.4 (−1.07)	−1.5 (−0.93)

Note: I = Inattention Presentation; H/I = Hyperactive/Impulsive Presentation; CG = Control group; C = Combined presentation. The values without parenthesis are the mean differences-*MD*; the values in parenthesis are the effect size values using the Cohen’s *d* statistic. * *p* < 0.05, *** < 0.005. Bonferroni correction = 0.008

## Data Availability

The data presented in this study are available on request from the corresponding author. The data are not publicly available due to ethical restrictions.

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
