# Peer review of "Effects of Sleep on the Academic Performance of Children with Attention Deficit and Hyperactivity Disorder"

_brainsci, 2021, doi:10.3390/brainsci11010097_

Round 1
Reviewer 1 Report
Dear authors
This is an impressive study evaluating sleep by actigraphic recording in a large sample of chuldne with ADHD and controls:
There some concerns that need to be addressed:
How are controls recruited?
Is this a prospective study?
You did not specify the assessment performed to controls. "Nearly all of these students (94%) had been 185 prescribed medication to control their ADHD symptoms)": since they were under treatment the aim of the study shoud change, and those without treatment analyzed separately.
Are the ADHD group under treatment during evaluation (actigraphic recording and cognitive tests), and still treated during weekends?
The manuscript is too long and too dispersive, the tables are too large with many data, not really readable.
In introduction please focus on studies and results of actigraphic recording in ADHD, and if there are studies after treatment. Please add some data about sleep phenotypes and about sleep delay in these children (that it is seems not the case of your population of Spanish children). Treatment should be specified.
Since you did not find any differences in sleep parameters (with the exception of Inattentive group and this is very interesting and in lines with the hypothesis that this is the hypersomnia- group of ADHD), your more interesting result is this, the normalization of sleep under treatment.
Author Response
The authors acknowledge the editor and reviewers their suggestions and comments on the manuscript and hope the changes we introduced will meet their expectations. These changes are described below, in response to the different aspects addressed in the revision. Also, manuscript was English revised by native speaker. The modifications in the manuscript have been highlighted using red colored text.Please let us know if you need additional information.
Reviewer 1
How are controls recruited?
- The process for recruiting the control group has been clarified in page 6 (lines 242-246).
Is this a prospective study?
- Thanks for your question, it has been added in the section of Design and Procedure the clarification that it is a cross-sectional study (page 6, lines 240-253).
You did not specify the assessment performed to controls. "Nearly all of these students (94%) had been 185 prescribed medication to control their ADHD symptoms)": since they were under treatment the aim of the study should change, and those without treatment analyzed separately.
- Thank you for your comment. We have clarified this aspect in the last paragraph of page 5.
Are the ADHD group under treatment during evaluation (actigraphic recording and cognitive tests), and still treated during weekends?
- According to your recommendation, it has been specified in the text that the subjects with pharmacological treatment maintained the treatment during the whole evaluation process (please see page 4, participants section).
The manuscript is too long and too dispersive, the tables are too large with many data, not really readable.
- Thanks for your suggestion. We have modified the tables making more readable, as well as, modified the text following the recommendations (please see again the Tables and read the different modifications carried out along the text).
In introduction please focus on studies and results of actigraphic recording in ADHD, and if there are studies after treatment. Please add some data about sleep phenotypes and about sleep delay in these children (that it is seems not the case of your population of Spanish children). Treatment should be specified.
- In accordance with this recommendation, a citation [32] has been added in allusion to this aspect, extending and justifying the study of the ADHD group with pharmacological treatment.
Since you did not find any differences in sleep parameters (with the exception of Inattentive group and this is very interesting and in lines with the hypothesis that this is the hypersomnia- group of ADHD), your more interesting result is this, the normalization of sleep under treatment.
- Thank you very much for your comment. Considering that we have not controlled the effect of pharmacological treatment, we cannot affirm this result. However, we have added this interesting idea as a future research line of our project. Please see future perspective and limitations section.
Reviewer 2 Report
In the present study, the authors are trying to find a correlation between sleep and academic performance of ADHD subjects compared to the controls. I think a few controls are missing and the manuscript needs a lot of work. I have a few suggestions and concerns.
- Rephrase line – 41.
- Might want to consider replacing the word “presentations”.
- Line 43 says ADHD-I is most common and line 47 says ADHD-C is most common. Which one is it?
- Do ADHD-I have no problems with memory? Is it specific to the C form or comes from I?
- Replace “conciliation” in line 68 with “consolidation”.
- Rephrase line 82.
- I don’t think the entire sleep paragraph in introduction is required. It could easily be shortened.
- Is sleep problem in any one type of ADHD?
- Rephrase lines 132-133.
- Aim section in line 157 – 162 is unnecessary and can be incorporated in lines 151-156.
- Unnecessary parenthesis in line 186.
- Why were controls not taken from independent schools?
- Were the controls tested for inattention and hyperactivity symptoms? I know they didn’t have the ADHD diagnosis.
- Grades pooled together and repeat correlations?
- How did they normalize grades from different schools – especially when controls are not from independent?
- What do lines 269-271 even mean?
- 307-308 – Please rephrase.
- How do they explain no difference in English in table 5?
- What do they think is the reason for no difference in sleep quality between ADHD and controls during the weekdays?
- Why have they not included a correlation with day time sleep?
Author Response
The authors acknowledge the editor and reviewers their suggestions and comments on the manuscript and hope the changes we introduced will meet their expectations. These changes are described below, in response to the different aspects addressed in the revision. Also, manuscript was English revised by native speaker. The modifications in the manuscript have been highlighted using red colored text.
Please let us know if you need additional information.
Reviewer 2
In the present study, the authors are trying to find a correlation between sleep and academic performance of ADHD subjects compared to the controls. I think a few controls are missing and the manuscript needs a lot of work. I have a few suggestions and concerns.
Rephrase line – 41.
- Thanks for the suggestion, the line has been slightly reformulated to make it clearer.
Might want to consider replacing the word “presentations”.
- Thanks for your comment. This study was carried out following DSM-5 (APA, 2013) and the scale administered to parents was also elaborated according to DSM criteria. For this reason, we consider that it is most precise mention “presentations” instead of “subtypes”. Moreover, if we only talked about “ADHD” in a general way, and without differentiating between presentations, possibly the conclusions will be more difficult to explain (and more difficult to extrapolate) considering the heterogeneity of this disorder.
Line 43 says ADHD-I is most common and line 47 says ADHD-C is most common. Which one is it?
- Thanks for your review, it has been modified the range lines 41-50 in order to clarify this aspect.
Do ADHD-I have no problems with memory? Is it specific to the C form or comes from I?
- Thanks, as indicated in the text, according to the DSM 5 manual, presentations are defined based on a series of characteristics, and the combined type presents alterations in memory to a greater degree than does the ADHD-I presentation. Please see first paragraph in page 2.
Replace “conciliation” in line 68 with “consolidation”.
- Done the modification suggested.
Rephrase line 82.
- Thank you for this suggestion, the line 82 has been rewritten.
I don’t think the entire sleep paragraph in introduction is required. It could easily be shortened.
- Thanks for your comment, we have modified the paragraph making more readable (please read again the section).
Is sleep problem in any one type of ADHD?
- Thank you, a new in text-citation has been added [30] in relation to your suggestion. This citation mentions that there are no differences in sleep within the presentations of ADHD.
Rephrase lines 132-133.
- Thanks, we have modified sleep and ADHD section.
Aim section in line 157 – 162 is unnecessary and can be incorporated in lines 151-156.
- Changes have been made following your suggestions.
Unnecessary parenthesis in line 186.
- Thanks, done.
Why were controls not taken from independent schools?
- The sampling method chosen was non-probabilistic casual or accessibility sampling for both groups used in this study. For clarifying this aspect, we have added additional information about recruitment process (please read the Design and Procedure section).
Were the controls tested for inattention and hyperactivity symptoms? I know they didn’t have the ADHD diagnosis.
- Yes, children from control group were administered the same scale used for the experimental group in order to check the absence of ADHD symptoms (please read the last paragraph of page 5).
Grades pooled together and repeat correlations?
- Table 2 showed the results from Pearson correlations between all variables analyzed in the present study. For that, we included the general values (it means, the general mean) for each variable analyzed. Please see again Preliminary analysis section.
How did they normalize grades from different schools – especially when controls are not from independent?
- Thank you. We have added this information in the second paragraph of page 7.
What do lines 269-271 even mean?
- Thank you, these lines have been deleted in order to make clearer the text.
307-308 – Please rephrase.
- Following the reviewer recommendation, the suggested lines has been reformulated. Please see the last paragraph of page 6.
How do they explain no difference in English in table 5?
- Regarding to academic performance in English, there are not statistically significant differences in relation to weekday bedtime variable, however if it is analyzed each subgroup (< 11 pm Vs. ≥ 11 pm), the combine presentation showed significant differences in comparison with its control group obtaining this one best results. We have added this information in the first paragraph of page 10.
What do they think is the reason for no difference in sleep quality between ADHD and controls during the weekdays?
- Thanks for your comment. The reason for no differences in sleep quality is partially explained by the school schedules, which make that sleep schedules remain within a normalized range during the weekdays, both in children with ADHD and without ADHD. Please read second paragraph of page 12.
Why have they not included a correlation with day time sleep?
- Thank you for your suggestion. It has been included as a limitation and future perspective of the study. It would be very interesting to study the effect of day time sleep for learning more about how the day rest affects academic performance. Please read last paragraph of page 12.
Round 2
Reviewer 2 Report
The edited manuscript looks good to me and my concerns have been addressed. I recommend publication of the modified manuscript.
Author Response
First of all, thanks for these comments and the time taken in this revision. We have highlighted the changes in yellow Your suggestions have significantly increased the quality of the present scientific paper.
- Table 3 is missing a notes section. What does * stand for? Also, the covariate columns are not clear. P-values not in parentheses, and partial eta in parentheses???? The coding for gender needs to be specified to work out which direction the effects go in.
- We have modified this aspect, the values without parentheses are effect size, while the values in parenthesis are p values (Lines 290-291). Regarding sex covariate, it is observed a tendency to present higher results in men than women (Lines 283-284). Please see again table 3.
- Similar problems with covariates in Table 4.
- We have added information about covariates. Please see again Table 4 (Lines 313-315).
- Please ensure that the text states how many participants were in each group when doing the bedtime breakdown analysis. If the numbers result in highly unequal sample sizes then perhaps be sure to emphasize in the results or discussion (limitations).
- Thanks for your suggestion. Participants of each groups has been include in the beginning of the sub-heading (Lines 330-331). The breakdown has been included in Table 5, which indicates the composition of each of the groups according to the weekend bedtime. In which a balance by groups and hours is evidenced. Also, we include the number by groups like limitation of the study (Lines 431-433).
- In the notes of Table 7, what is the measure of effect size (Cohen’s D)?
- Thanks for your comment, the values without parenthesis are the Mean Differences-MD; the values in parenthesis are the effect size values using the Cohen’s d statistic (Lines 383-385). Please see again Table 7.
- Avoid the use of the word “subjects”, use “participants” instead.
- We have replaced the term “subjects” by “participants” (Please read again the text). Only we have kept the term “subject” for referring to the “inter-subject effect” because is a statistical expression.
- Line 259: AHDH vs ADHD.
- We have checked this mistake (line 260).